# Alleviation of Aflatoxin B1-Induced Hepatic Damage by Propolis: Effects on Inflammation, Apoptosis, and Cytochrome P450 Enzyme Expression

**DOI:** 10.3390/cimb48010056

**Published:** 2026-01-01

**Authors:** Sevtap Kabalı, Neslihan Öner, Ayca Kara, Mehtap Ünlü Söğüt, Zehra Elgün

**Affiliations:** 1Department of Nutrition and Dietetics, Faculty of Health Sciences, Ondokuz Mayıs University, 55020 Samsun, Türkiye; sevtap.kkurtaran@omu.edu.tr (S.K.); mehtap.sogut@omu.edu.tr (M.Ü.S.); 2Department of Nutrition and Dietetics, Faculty of Health Sciences, Erciyes University, 38039 Kayseri, Türkiye; zavanoglu@erciyes.edu.tr; 3Betül Ziya Eren Genom and Stem Cell Center (GENKOK), Erciyes University, 38280 Kayseri, Türkiye

**Keywords:** aflatoxin B1, propolis, liver, cytochrome P450 enzyme system, oxidative stress, apoptosis, histopathology, immunohistochemistry, inflammation, propolis

## Abstract

*Aflatoxin**B1* (AFB1) is a hepatotoxic mycotoxin whose bioactivation by cytochrome P450 (CYP450) enzymes generates reactive metabolites that drive oxidative stress, inflammation, and apoptosis. Propolis is a bee-derived product with antioxidant and immunomodulatory properties. To investigate whether propolis supplementation attenuates AFB1-induced hepatic injury by modulating inflammatory mediators, Nrf2–HO-1 signaling, mitochondrial apoptosis, and CYP450 expression in rats, twenty-four male Sprague-Dawley rats were randomly allocated to four groups (n = 6): control, AFB1 (25 µg/kg/day), propolis (250 mg/kg/day), and AFB1 + propolis. Treatments were given by oral gavage for 28 days. Hepatic IL-1β, IL-6, TNF-α, Nrf2 and HO-1 levels were measured by ELISA. Histopathology was assessed on H&E-stained sections. Bax, Bcl-2, caspase-3, CYP1A2, CYP3A4, CYP2C19 and cytochrome P450 reductase expressions were evaluated immunohistochemically and quantified by ImageJ. Data were analyzed using one-way ANOVA with Tukey’s post hoc test. AFB1 significantly increased hepatic IL-1β and IL-6 and reduced Nrf2 levels, while propolis supplementation restored Nrf2, elevated HO-1 and significantly lowered IL-6 compared with AFB1 alone (*p* < 0.05). AFB1 induced marked hydropic degeneration, sinusoidal congestion, and mononuclear infiltration, alongside increased Bax and caspase-3 and decreased Bcl-2 expression; these changes were largely reversed in propolis-treated groups. AFB1 upregulated CYP1A2, CYP3A4 and cytochrome P450 reductase, whereas propolis co-treatment significantly suppressed their expression without affecting CYP2C19. Propolis supplementation attenuated AFB1-induced liver injury through coordinated anti-inflammatory, antioxidant, anti-apoptotic and metabolic regulatory effects, notably via restoration of Nrf2–HO-1 signaling and down-regulation of key CYP450 isoenzymes. Propolis may represent a promising natural dietary strategy against AFB1-associated hepatotoxicity, warranting further translational research.

## 1. Introduction

*Aflatoxin B1* (AFB1), a highly potent mycotoxin produced primarily by *Aspergillus flavus* and *Aspergillus parasiticus*, represents a major global food safety concern due to its pronounced hepatotoxic, immunosuppressive, genotoxic, and carcinogenic effects [1,2,3,4]. Contamination occurs predominantly in staple crops such as maize, peanuts, and various cereals, particularly in hot and humid climates that promote fungal proliferation [2]. The International Agency for Research on Cancer (IARC) classifies AFB1 as a Group 1 carcinogen and highlights its strong association with hepatocellular carcinoma development [3]. Regions with insufficient food storage and monitoring systems bear the greatest risk, resulting in substantial public health and economic consequences [5,6,7].

The toxicological actions of AFB1 arise from interconnected mechanisms involving oxidative stress, inflammation, mitochondrial dysfunction, and apoptosis, all of which contribute to progressive hepatic injury [4,5,8]. Bioactivation of AFB1 by cytochrome P450 enzymes—particularly CYP1A2 and CYP3A4—leads to the formation of the highly reactive AFB1-8,9-epoxide, which induces lipid peroxidation, reactive oxygen species (ROS) generation, and DNA adduct formation [4,8]. These molecular events undermine hepatocyte structural integrity and metabolic function, resulting in both acute aflatoxicosis—characterized by hemorrhagic necrosis, bile duct proliferation, and edema—and chronic outcomes such as hepatocellular carcinoma [7,8,9].

Exposure to aflatoxins, particularly in regions with inadequate food storage practices, leads to chronic health issues and economic losses [6]. In animals, aflatoxins cause liver damage, decreased milk production, reduced reproductive capacity, and suppressed immunity, even at low dietary concentrations [7]. The primary target organ for both acute and chronic aflatoxin injury is the liver, where high doses over short periods can cause acute aflatoxicosis, characterized by hemorrhagic necrosis, bile duct proliferation, edema, and lethargy. Meanwhile, chronic exposure to lower doses may lead to hepatocellular carcinoma [8]. Due to these severe health implications, the presence of aflatoxins in food products is regulated in many countries, with the European Union setting a limit of 2 μg/kg in foods for direct human consumption [9]. Given the limitations of synthetic preservatives and their associated toxic effects, there is a growing interest in exploring natural alternatives for food preservation and protection against aflatoxin-induced damage [4].

Propolis, a resinous substance collected from plant exudates by honeybees, has attracted great interest due to its antioxidant, anti-inflammatory, hepatoprotective, and anticancer properties. Its bioactive compounds, including flavonoids, phenolic acids, and terpenes, contribute to free radical scavenging activity by reducing oxidative stress-induced damage [10]. Studies have shown that propolis supplementation can increase antioxidant enzyme activities (superoxide dismutase, catalase, etc.), reduce lipid peroxidation, and protect against DNA damage in various toxicological models, including aflatoxin exposure [11,12].

Furthermore, propolis exhibits immunomodulatory effects that may help counteract AFB1-induced immunosuppression [13]. It has been shown to regulate inflammatory cytokines, inhibit apoptosis in hepatocytes, and promote liver regeneration following toxicant exposure [14,15]. These findings suggest that propolis may serve as a natural dietary intervention to reduce the deleterious effects of AFB1, especially in high-exposure risk areas. Although propolis has been studied for its potential to reduce oxidative damage and prevent liver tissue injury, no research has specifically addressed its role in preventing immune and liver tissue damage, as well as mitigating inflammation and apoptosis caused by AFB1 exposure. Thus, some research questions were identified for this research:RQ1. How does propolis supplementation influence oxidative stress and inflammation in AFB1-induced liver damage?RQ2. Can propolis mitigate AFB1-induced apoptosis in hepatocytes and promote liver regeneration?RQ3. What are the potential immunomodulatory effects of propolis in counteracting AFB1-induced immunosuppression?RQ4. How effective is propolis as a natural dietary intervention in reducing the toxicological impact of AFB1 exposure?

This study aimed to investigate the effects of AFB1 on pro-inflammatory parameters, apoptosis, liver tissue damage, and expression of cytochrome P450 (CYP450) enzymes, as well as to evaluate the protective role of propolis supplementation in mitigating AFB1-induced metabolic damage.

## 2. Materials and Methods

### 2.1. Experimental Animals

In this study, 24 adult male Sprague-Dawley albino rats aged 6–8 weeks (200–250 g) were used. The ARRIVE guidelines and the 3R principles (Replacement, Reduction, and Refinement) were strictly adhered to, ensuring animal welfare and compliance with ethical standards. All animals were randomly assigned to cages (two rats per cage) with ad libitum access to a standard pellet diet and tap water. Environmental conditions were controlled to maintain a constant temperature (21 ± 1 °C) and humidity (55 ± 10%) under a 12 h light/dark cycle. The study was approved by the Ondokuz Mayıs University Local Ethics Committee for Animal Experiments (Decision No. 2023/55).

### 2.2. Preparation and Dosage of AFB1

AFB1 (5 mg, ≥99.5%) (CAS No: 1162-65-8), in powder form, was obtained from CAYMAN Chemical Company (Ann Arbor, MI, USA) and dissolved in dimethyl sulfoxide (DMSO) (5 mL, ≥99.5%) (CAS No: 67-68-5, Sigma-Aldrich Chemical Company, St. Louis, MO, USA) due to its minimal effects on animals [16]. Previous studies have reported that daily administration of AFB1 at a dose of 25 µg/kg body weight for four weeks induces toxic effects in rats [16,17,18]. Furthermore, oral gavage was chosen as the route of administration due to challenges in achieving homogeneous distribution of AFB1 in feed, variability in individual food intake, and difficulties in dose control. Based on these findings, AFB1 was administered by oral gavage at a daily dose of 25 µg/kg for four weeks in this study. This dosage corresponds to AFB1 concentrations ranging from 0.03 to 0.45 mg/kg (30–450 ppb) in food, which have been reported as exposure levels among individuals in developing countries.

### 2.3. Extraction and Dosage of Propolis

Brown raw propolis (*Populus* sp.), collected from Samsun, Turkey, was extracted for use in this study. Propolis extracts were prepared following the protocol described by Cai et al. [19]. The results of liquid chromatography–mass spectrometry (LC–MS/MS) analysis of the extracted propolis are presented in Appendix A. Previous studies have shown that propolis supplementation at a daily dose of 250 mg/kg reduces inflammation and tissue damage in rats. Based on these findings, propolis was administered by oral gavage at a daily dose of 250 mg/kg for four weeks in this study [20,21]. Experimental studies on animals have indicated that the safe dose of propolis for humans is approximately 70 mg/day [22]. The dose administered in this study was calculated according to the human–rat dose conversion and was determined to correspond to approximately 8.1 mg/day in humans.

### 2.4. Experimental Groups

The rats were randomly divided into four experimental groups, each consisting of six animals. These groups were organized as follows: Control group (administered diluted DMSO), AFB1 group (AFB1 25 µg/kg/day oral gavage), propolis group (propolis 250 mg/kg/day oral gavage) and AFB1 (AFB1 25 µg/kg/day oral gavage) + propolis group (propolis 250 mg/kg/day oral gavage). All treatments were administered daily for 28 days. The number of animals used was determined by power analysis (α = 0.05, β = 0.95) [23]. Sample size calculation was performed using G*Power 3.1.9.7 software (Düsseldorf, Germany). A one-week adaptation period was applied prior to the experiment to monitor the animals’ health status and reduce stress. The experimental setup is illustrated in Figure 1.

At the end of the treatment period, the rats were euthanized by decapitation under deep anesthesia. Liver tissues were immediately dissected for biochemical, histological, and immunohistochemical analyses.

### 2.5. Enzyme-Linked Immunosorbent Assay (ELISA) Measurements

In this study, inflammatory markers, including interleukin-1β (IL-1β; Cat. No: E0119Ra), interleukin-6 (IL-6; Cat. No: E0135Ra), and tumor necrosis factor-α (TNF-α; Cat. No: E0764Ra), were analyzed to assess inflammation in liver tissue. Additionally, levels of nuclear factor erythroid 2-related factor 2 (Nrf2; Cat. No: E1083Ra) and heme oxygenase-1 (HO-1; Cat. No: E0676Ra) were measured using ELISA kits obtained from BT-LAB (Shanghai, China).

### 2.6. Histological Staining

The dissected liver tissues were fixed in 10% formalin. Following dehydration and clearing with xylene, the tissues were embedded in paraffin and blocked. Serial sections of 5 μm thickness were obtained from the paraffin blocks and mounted onto slides. The paraffin was removed from the sections using standard histological procedures. To assess the general histological architecture, tissue sections were stained with hematoxylin and eosin (H&E), and liver morphology was evaluated under a light microscope by an experienced histologist who was blinded to the experimental groups (Leica DM500, Wetzlar, Germany).

### 2.7. Immunohistochemistry

For immunohistochemical analysis, 5 μm serial sections were obtained from paraffin blocks and mounted onto polylysine-coated slides. Polyclonal antibodies against Bcl-2 (Cat. No: E-AB-60012), Bax (Cat. No: E-AB-10049), caspase-3 (Cat. No: E-AB-60017), CYP1A2 (Cat. No: 19936-1-AP), CYP2C19 (Cat. No: PA5-114368), CYP3A4 (Cat. No: 18227-1-AP), and Cytochrome CYP450 Reductase (Cat. No: PA5-117120) were used. Immunohistochemical staining was performed according to the manufacturer’s instructions (Thermo Fisher, Cheshire, UK) following antibody optimization. The stained sections were photographed using a Leica microscope at 10× magnification, and the intensity of antibody expression was quantified using the ImageJ software (Version 1.8.0_112, Bethesda, MD, USA).

### 2.8. Statistical Analysis

Statistical analyses were performed using GraphPad Prism software (Version 10.0, San Diego, CA, USA). The normality of data distribution was assessed using the Kolmogorov–Smirnov test. Results were expressed as mean ± standard deviation (SD). Comparisons between groups were conducted using one-way analysis of variance (ANOVA), followed by Tukey’s post hoc test for variables showing statistically significant differences. A *p*-value of <0.05 was considered statistically significant.

## 3. Results

### 3.1. IL-1β, IL-6, TNF-α, Nrf2, and HO-1 Levels

As shown in Figure 2, IL-1β and IL-6 levels were significantly higher in the AFB1 group compared with both the control and propolis groups (IL-1β: Control, 2.15 ± 0.29 ng/g; AFB1, 2.65 ± 0.28 ng/g; Propolis, 2.11 ± 0.43 ng/g. IL-6: Control, 2.21 ± 0.25 ng/g; AFB1, 2.85 ± 0.39 ng/g; Propolis, 2.25 ± 0.34 ng/g; *p* < 0.05). Figure 2 also demonstrates that IL-6 levels were significantly lower in the AFB1 + Propolis group compared with the AFB1 group (2.31 ± 0.32 ng/g; *p* < 0.05). Although TNF-α levels tended to increase in the AFB1 group and decrease in the propolis-treated groups, no statistically significant differences were observed among the groups.

Figure 2 further shows that Nrf2 levels were significantly lower in the AFB1 group than in all other groups (Control, 3.54 ± 0.18 ng/g; AFB1, 3.13 ± 0.26 ng/g; Propolis, 3.72 ± 0.34 ng/g; AFB1 + Propolis, 3.67 ± 0.27 ng/g; *p* < 0.05). HO-1 levels were significantly higher in the propolis group compared to both the control and AFB1 groups (Control, 2.49 ± 0.40 ng/g; AFB1, 2.45 ± 0.19 ng/g; Propolis, 3.02 ± 0.35 ng/g; AFB1 + Propolis, 2.96 ± 0.34 ng/g; *p* < 0.05).

### 3.2. Histological Investigation

According to the results of H&E staining, hepatocytes in the control group maintain standard histological architecture. In the AFB1 group, dense hydropic degeneration and radial cell alignment are observed in the portal areas. Vacuolar spaces and granular structures in the cell cytoplasm affect the morphological appearance of the cells. Additionally, sinusoidal expansion, congestion, and widespread mononuclear cell infiltration are observed. In the propolis group, cells exhibit histomorphological features similar to those in the control group. In the AFB1 + Propolis group, a decrease in hydropic degeneration, narrowing of sinusoidal spaces, and a decrease in vacuolar appearance are noted compared to the AFB1 group. However, mononuclear cell infiltration was still observed but was less intense than in the AFB1 group (Figure 3). These findings suggest that AFB1 exposure negatively affects the histological structure of hepatocytes, while propolis supplementation may help preserve the histological integrity of liver tissue. These histological observations agree with the biochemical and immunohistochemical data, in which AFB1 exposure was associated with elevated IL-6, Bax and Casp-3 levels and reduced Bcl-2 and Nrf2, whereas propolis supplementation largely normalized these parameters and paralleled the improvement in tissue architecture.

### 3.3. Immunohistochemical Investigation

To examine the effects of AFB1 exposure and propolis supplementation on the expression of apoptosis-related proteins in rat liver tissue (Figure 4). Bax expression level was found to be higher in the AFB1 group compared to the other groups (*p* < 0.01). Similarly, Cas-3 expression level was higher in the AFB1 group than in the control and propolis groups (*p* < 0.01). On the contrary, the Bcl-2 expression level of the AFB1 group was lower than the propolis-supplemented groups (Propolis and AFB1 + Propolis) (*p* < 0.05).

The expression levels of CYP2c19, CYP3c4, CYP1c2, and Cytochrome P450 reductase enzymes of the experimental groups were examined using the immunohistochemical method (Figure 5). There was no statistically significant difference between the groups in terms of the expression levels of CYP2c19 enzyme (*p* > 0.05). It was observed that the expression levels of CYP3c4, CYP1c2, and Cytochrome P450 reductase enzymes were increased in the AFB1 group compared to the control group (*p* < 0.01); the propolis group and the control group were similar (*p* > 0.05). When AFB1-exposed rats were supplemented with propolis, the expression levels of these enzymes were found to be lower compared to the AFB1 group (*p* < 0.01).

## 4. Discussion

In this study, we investigated the hepatic impact of AFB1 exposure on inflammation, apoptosis, histological integrity, and CYP450 enzyme expression, as well as the potential protective role of propolis supplementation. As anticipated, AFB1 administration induced a pronounced pro-inflammatory response, disrupted hepatic histological architecture, enhanced the expression of pro-apoptotic markers, and upregulated the expression of CYP3c4, CYP1c2, and cytochrome P450 reductase. Propolis supplementation effectively mitigated these adverse effects by reducing IL-6 levels, restoring Nrf2–HO-1 signaling, re-establishing the Bax/Bcl-2/Cas-3 balance towards cell survival, and attenuating CYP450 overexpression. Collectively, these findings suggest that propolis alleviates AFB1-induced hepatic injury through a combination of anti-inflammatory, antioxidant, anti-apoptotic, and metabolic regulatory mechanisms.

Our results confirm that administration of AFB1 at a dose of 25 μg/kg for 28 days triggered a significant pathological response in the liver. This response included: increased pro-inflammatory cytokines (IL-1β, IL-6), suppression of protective antioxidant pathways (Nrf2), activation of the pro-apoptotic cascade (Bax, Caspase-3), and characteristic histopathological lesions (hydropic degeneration, mononuclear infiltration). We demonstrated that the protective mechanism of propolis involves a multifaceted cellular defense strategy, extending beyond its role as a free radical scavenger. This strategy includes: (i) powerfully activating the endogenous antioxidant defense system (Nrf2/HO-1), (ii) suppressing pro-inflammatory cytokine production, (iii) inhibiting the mitochondrial apoptosis cascade, and (iv), most importantly, suppressing the expression of key CYP450 enzymes (CYP1c2, CYP3c4) responsible for the toxic bioactivation.

In our study, the significant increase in IL-1β and IL-6 levels in the AFB1 group compared to the control group aligns with previous studies [24,25,26,27], which report that AFB1 activates an NF-κB-mediated pro-inflammatory response in liver tissue. In particular, it has been reported that AFB1 increases the secretion of pro-inflammatory cytokines such as IL-1β and IL-6 by regulating the inflammatory microenvironment in hepatocytes and activating the NF-κB signaling pathway [27,28]. It is well established that oxidative stress induced by AFB1 contributes to the overexpression of these cytokines by accelerating the degradation of IκB and the nuclear translocation of NF-κB [24,29,30]. It has been demonstrated that the production of reactive oxygen species induced by AFB1 activates the NF-κB signaling pathway, leading to the upregulation of pro-inflammatory cytokines [24].

Both experimental models and human studies have reported that propolis and its phenolic compounds (particularly caffeic acid phenethyl ester, quercetin, and quercetin derivatives) inhibit the NF-κB pathway, thereby reducing pro-inflammatory cytokines such as IL-1β, IL-6 and, to a lesser extent, TNF-α [31,32,33,34,35,36]. In our study, AFB1 exposure significantly increased IL-1β and IL-6 levels compared to the control group. In contrast, propolis supplementation significantly lowered IL-6 concentrations in the AFB1 + propolis group, producing a non-significant downward trend in IL-1β. In this context, the significant reduction in IL-6, together with the tendency toward lower IL-1β levels, can be attributed to the antioxidant capacity of propolis, coupled with its ability to modulate NF-κB signaling and mitigate the AFB1-triggered inflammatory response [31,35]. However, the lack of a statistically significant difference in TNF-α levels suggests that AFB1’s effect on this specific cytokine might be weaker or more time-dependent, and that the regulatory impact of propolis on the overall cytokine profile is more pronounced for IL-1β and IL-6.

In line with these mechanisms, our study demonstrated that Nrf2 levels were significantly lower in the AFB1 group than in all other groups, whereas both propolis and AFB1 + propolis supplementation restored Nrf2 levels to, or slightly above, control values. HO-1 concentrations were particularly elevated in the propolis-only group, indicating a strong activation of downstream antioxidant defenses. These findings are consistent with previous reports showing that propolis flavonoids, especially caffeic acid phenethyl ester and quercetin derivatives, dissociate Nrf2 from Keap1, facilitate its nuclear translocation, and induce the expression of antioxidant and detoxifying enzymes such as HO-1 [36,37]. Through this mechanism, Nrf2 activation not only enhances cellular redox control but also indirectly dampens NF-κB-driven inflammation by reducing ROS production and limiting the transcription of pro-inflammatory genes [35,37,38,39]. Thus, the concurrent restoration of Nrf2/HO-1 signaling and the attenuation of IL-6 levels in propolis-treated groups suggest that the hepatoprotective effect of propolis involves a coordinated regulation of oxidative stress and inflammatory pathways.

Mechanisms exist by which propolis-mediated Nrf2 activation not only enhances antioxidant defense but also plays a role in regulating inflammatory pathways [34,35,38]. A robust and intricate crosstalk exists between the Nrf2 and NF-κB pathways [36]. Activation of Nrf2 restores cellular redox balance, thereby decreasing the production of reactive oxygen species [35,39]. This, in turn, leads to a decrease in inflammatory signals and indirectly dampens NF-κB activation [40]. For instance, caffeic acid phenethyl ester, a significant component of propolis, has been shown to dissociate Nrf2 from Keap1, facilitating its nuclear translocation and thereby inducing the expression of antioxidant and detoxification enzymes, such as HO-1 [36,37]. Furthermore, Nrf2 activators have been reported to exert anti-inflammatory effects by suppressing the activation of NF-κB and the expression of its target genes [40].

This dual mechanism suggests that propolis’s reduction of IL-1β and IL-6 levels occurs not solely through direct NF-κB inhibition [41,42], but also via the Nrf2-mediated restoration of redox homeostasis [34,35]. Propolis simultaneously suppresses both oxidative stress and inflammation caused by toxicity through two distinct mechanisms: reducing ROS production and directly inhibiting NF-κB, thereby creating a more favorable cellular environment for hepatocytes [38,42]. The changes in Bax, Bcl-2, and caspase-3 expression suggest activation of apoptotic signaling, although direct confirmation would require analysis of cleaved caspase-3 or PARP. Although AFB1 exposure increased Bax and caspase-3 expression and reduced Bcl-2, these immunohistochemical changes alone do not conclusively confirm apoptosis, as cleaved caspase-3 or cleaved PARP were not assessed. Therefore, the observed alterations should be interpreted as activation of apoptotic signaling rather than definitive evidence of apoptosis.

Histopathological examination clearly revealed severe signs of cellular damage in the liver tissue of the AFB1 group. These damages included intense hydropic degeneration, vacuolar spaces in the cytoplasm, and granular structures within hepatocytes [43,44]. Hydropic degeneration is one of the first manifestations of cellular damage, serving as a serious indicator of cellular injury and metabolic disturbance, characterized by intracellular water accumulation resulting from imbalances in the cell’s ability to maintain ionic balance and fluid homeostasis [4]. Furthermore, sinusoidal enlargement, congestion, and widespread mononuclear cell infiltration were observed in the hepatic parenchyma [44]. Sinusoidal congestion and inflammatory cell infiltration indicate a high acute inflammatory response and vascular stress, suggesting a process that may potentially progress to necrosis [45]. These findings confirm that AFB1 not only exerts direct toxic effects on hepatocytes but also severely impairs the liver’s microcirculation and immune response [45].

In the present study, immunohistochemical analyses demonstrated that AFB1 exposure results in changes consistent with the activation of the mitochondrial apoptotic pathway. There is a well-established crosstalk between the Nrf2 and NF-κB signaling systems, whereby suppression of Nrf2 enhances ROS formation, promotes IκB degradation and facilitates NF-κB activation. This ultimately increases IL-1β and IL-6 production [24,29,35,38]. In line with previous reports demonstrating ROS-dependent NF-κB activation under AFB1 exposure, our study observed elevated IL-1β and IL-6 levels in the AFB1 group.

Propolis supplementation restored Nrf2 expression and increased HO-1 levels, consistent with previous findings that propolis flavonoids (e.g., CAPE and quercetin derivatives) promote Nrf2 nuclear translocation and upregulate antioxidant defense genes [36,37]. Activation of the Nrf2/HO-1 axis reestablishes intracellular redox homeostasis and indirectly suppresses NF-κB-driven cytokine transcription by reducing oxidative stress and limiting pro-inflammatory signaling [35,37,38,39]. These mechanisms are consistent with prior evidence indicating that Nrf2 activation mitigates hepatic inflammation by downregulating NF-κB target genes [40].

The normalization of Bax, Bcl-2 and Casp-3 expression in propolis-treated groups also supports earlier reports showing that reductions in oxidative damage decrease p53 activation and mitochondrial permeabilization, thereby shifting the Bax/Bcl-2 ratio towards cell survival [31,32,33]. Although our findings suggest the activation of apoptotic signaling, direct biochemical confirmation would require the measurement of cleaved caspase-3 or PARP.

The AFB1-driven upregulation of CYP1a2 and CYP3a4 observed in our study is consistent with established mechanisms, indicating that these enzymes catalyze the formation of the toxic AFB1-8,9-epoxide. This promotes DNA adduct formation and hepatocarcinogenicity [4,8]. The absence of changes in CYP2c19 expression, combined with the selective suppression of CYP1a2, CYP3a4 and P450 reductase by propolis, suggests that propolis modulates specific metabolic bioactivation pathways rather than inhibiting CYP450 activity globally [10,11]. This selective downregulation is in line with previous studies that have reported propolis phenolics to attenuate CYP-mediated toxin activation [31,36]. Importantly, CYP2C19 expression remained unchanged across all groups, indicating that AFB1 does not uniformly modulate all hepatic CYP isoforms. This stability suggests that CYP1a2 and CYP3a4 are the principal enzymes involved in AFB1 bioactivation in rat liver, while CYP2c19 appears to play a minimal or non-responsive role under these experimental conditions.

Taken together, the above literature-supported mechanisms corroborate our findings and confirm that the hepatoprotective effects of propolis arise through coordinated antioxidant, anti-inflammatory, anti-apoptotic and metabolic regulatory actions.

This study has some limitations. First, only male Sprague-Dawley rats were used, and a single dose of AFB1 and propolis was evaluated over a 28-day period; differences between species at different doses and durations were not investigated. Second, the immunomodulatory effect was evaluated only through hepatic cytokine levels; the systemic immune response and immune organs (e.g., spleen) were not examined. Third, the propolis sample used was obtained from a specific geographical region (Samsun, Turkey), and its composition may vary between samples. Considering these limitations, further studies involving different doses, durations, and populations are needed to generalize our findings.

## 5. Conclusions

In conclusion, chronic exposure to AFB1 at a dose of 25 µg/kg for 28 days induced marked hepatic inflammation, oxidative stress, mitochondrial apoptosis, histological injury and up-regulation of CYP1a2, CYP3a4 and cytochrome P450 reductase in rats. Propolis supplementation at 250 mg/kg partially counteracted these changes by restoring Nrf2–HO-1 signaling, significantly reducing IL-6 concentrations, normalizing Bax/Bcl-2/Casp-3 expression and attenuating CYP450 overexpression, which together translated into a clear improvement in liver histology. These data support the hepatoprotective potential of propolis against AFB1-induced liver damage, although extrapolation to humans requires caution.

## Figures and Tables

**Figure 1 cimb-48-00056-f001:**
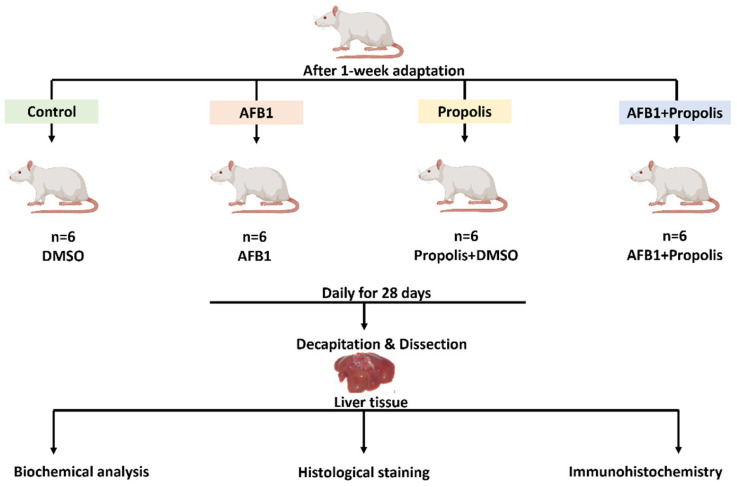
Experiment design. AFB1: Aflatoxin B1, DMSO: dimethyl sulfoxide.

**Figure 2 cimb-48-00056-f002:**
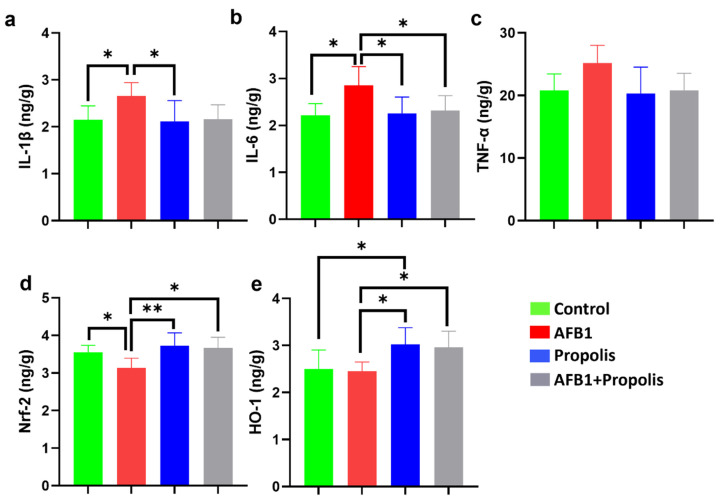
Evaluation of various biochemical parameters in liver tissues. IL-1β: Interleukin-1β (**a**), IL-6: Interleukin-6 (**b**), TNF-α: Tumor necrosis factor-α (**c**), Nrf2: Nuclear factor erythroid 2-related factor (**d**), HO-1: Heme oxygenase-1 (**e**). Data are expressed as mean ± SD (n = 6 per group). One-way ANOVA followed by Tukey’s post hoc test was used to analyze the data. * *p* < 0.05, ** *p* < 0.001 indicates statistical significance. TNF-α levels showed no significant differences among the groups.

**Figure 3 cimb-48-00056-f003:**
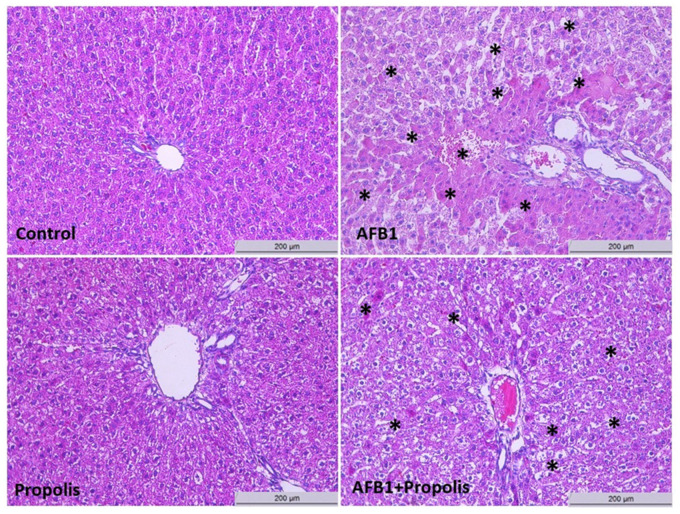
Histological evaluation of liver tissues in experimental groups (H&E, 20×). The control group exhibits normal hepatic histological architecture. In the AFBI group, pronounced hydropic degeneration, cytoplasmic vacuolisation, sinusoidal dilation and congestion, and marked mononuclear cell infiltration are observed in portal areas. The Propolis group shows histomorphological features comparable to the control group. In the AFBI-Propolis group, hydropic degeneration and vacuolisation are attenuated, and sinusoidal spaces appear narrowed compared to the AFBI group, while mononuclear cell infiltration is reduced but still present. Asterisks (*) indicate histopathological alterations, including hydropic degeneration, vacuolar degeneration, sinusoidal congestion, and mononuclear cell infiltration.

**Figure 4 cimb-48-00056-f004:**
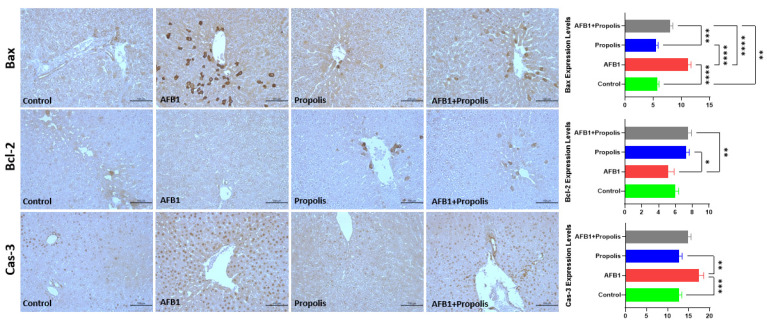
Immunohistochemical staining of Bax, Bcl-2 and Cas-3 proteins in liver tissues of the experimental groups and immunoreactivities of these proteins. Bax: Bcl-2-associated X protein, Bcl-2: B-cell lymphoma 2, Cas-3: Cysteine-aspartic acid protease 3. Scale bar: 200 μm, Objective: 10×. Data are expressed as the mean ± SD (n = 6 in per group). A one-way ANOVA test followed by a Tukey post hoc test was used to analyze the data. * *p* < 0.05, ** *p* < 0.01, *** *p* < 0.001, **** *p* < 0.0001.

**Figure 5 cimb-48-00056-f005:**
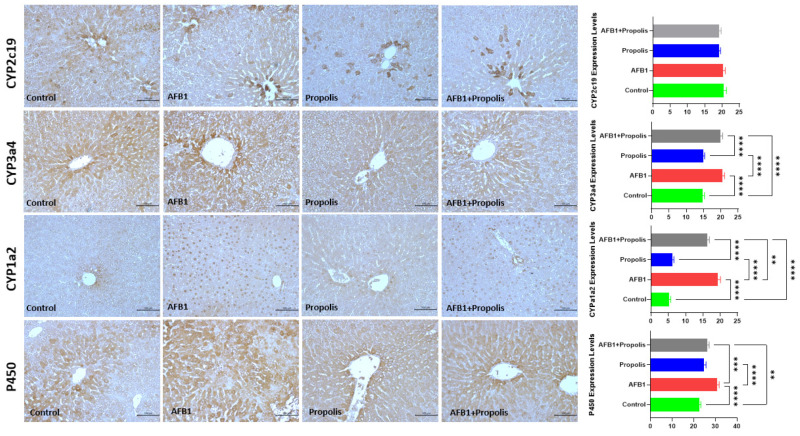
Immunohistochemical staining of CYP2c19, CYP3c4, CYP1c2, and Cytochrome P450 reductase enzymes in liver tissues of the experimental groups and immunoreactivities of these enzymes. CYP2c19: Cytochrome P450 family 2 subfamily C member 19, CYP3c4: Cytochrome P450 family 3 subfamily A member 4, CYP1c2: Cytochrome P450 family 1 subfamily A member 2. Scale bar: 200 μm, Objective: 10×. Data are expressed as the mean ± SD (n = 6 per group). A one-way ANOVA test followed by a Tukey post hoc test was used to analyze the data. ** *p* < 0.01, *** *p* < 0.001, **** *p* < 0.0001.

## Data Availability

The original contributions presented in this study are included in the article/Appendix A. Further inquiries can be directed to the corresponding authors.

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
