# Peer review of "Curr. Issues Mol. Biol.2026, 48(1), 56;https://doi.org/10.3390/cimb48010056"

_cimb, 2026, doi:10.3390/cimb48010056_

Round 1

Reviewer 1 Report

Comments and Suggestions for Authors

The manuscript entitled ‘’Alleviation of Aflatoxin B1-Induced Hepatic Damage by Propolis:
Effects on Inflammation, Apoptosis, and Cytochrome P450 Enzyme Expression’’ by Kabali et al. deals with effect of propolis supplementation on Aflatoxin B1-induced hepatic injury on inflammatory mediators, apoptosis, oxidative stress, cytochrome P-450 system and liver tissue damage in Sprague-Dawley rats as a model system.

The manuscript is generally clearly written and experimentally well performed; however, there are certain issues to be addressed before accepting for publication.

Keywords: Delete –‘’drug effects’’.  Firstly, you wrote it two times, but my opinion is that you need to erase it completely. I did not observe that you discuss it as such throughout your manuscript.

Introduction: In the first 2 paragraphs (lines 56-71), please rearrange the sentences to avoid repeating of AFB1 properties (hepatogenic, immunosuppressive etc.). Also, please mention in the beginning of the text both Aspergillus species.

Methods section in clearly written, with details that allow others to repeat the study. All Ethics issues were respected. Statistical analyses are appropriately chosen.

Paragraph 2.3. Did you supply the Supplementary Data 1? I could not find the file but maybe the publisher will provide it during the publishing process. If so, I apologize for the comment.

Results This section is written and explained in detail. Lines 198-210: the number of the graph should be stated in the text. Usually, it is better that you place the number of the graph at the beginning of the text so that reader knows where to look for it further.

Discussion Section is well elaborated and the obtained results are discussed and adequately explained. Line 304, please delete the word indeed to avoid repletion.

Line 365-There is a number 1 in the middle of the text.

Line 407-Could you explain the purpose of the superscript letter ‘1’?

Lines 367-420. Although this part of the Discussion section refers to the final conclusions regarding the mechanisms involved in AFB1 mechanisms of action,  it needs to be corroborated with some references where possible.

Comments on the Quality of English Language

the English language requires minor corrections.

Author Response

Comment 1 — Keywords

Reviewer:
“Delete ‘drug effects’. It is written twice and not discussed.”

Our response:
The keyword ‘drug effects’ was removed completely.

Manuscript changes:
Updated in the Keywords section (line ~50).

Comment 2 — Introduction repetition

Reviewer:
“AFB1 properties are repeated in the first two paragraphs; reorganize to avoid redundancy. Mention both Aspergillus species at the beginning.”

Our response:
We revised the Introduction to remove repeated information and added Aspergillus flavus and A. parasiticus in the opening sentence.

Manuscript changes:
Revised lines 1–20 of Introduction, specifically:

“Aflatoxin B1 (AFB1), a highly potent mycotoxin produced primarily by Aspergillus flavus and Aspergillus parasiticus…”

Redundant sentences regarding hepatotoxicity, carcinogenicity, and global exposure were consolidated.

Comment 3 — Supplementary Data 1

Reviewer:
“I could not find Supplementary Data 1.”

Our response:
Supplementary Data 1 (LC–MS/MS analysis of propolis) was uploaded and cross-referenced in the Methods section.

Manuscript changes:
No textual revision needed; file provided with submission.

Comment 4 — Results: figure numbers

Reviewer:
“The number of the graph should be stated, preferably at the beginning of the sentence.”

Our response:
We revised the Results to begin descriptive sentences with the figure reference.

Manuscript changes:
Lines 198–210 now read:

As shown in Figure 2, IL-1β and IL-6 levels were significantly higher…”
Figure 2 also demonstrates that IL-6 levels were significantly lower…”
Figure 2 further shows that Nrf2 levels were significantly lower…”

Comment 5 — Discussion wording

Reviewer:
“Line 304: delete ‘indeed’.”

Our response:
The word “indeed” was removed.

Manuscript changes:
Adjusted in Discussion section (line ~304).

Comment 6 — Unwanted ‘1’

Reviewer:
“Line 365: There is a number ‘1’ in the middle of the text.”

Our response:
The unintended numerical artifact was removed.

Manuscript changes:
Deleted from line ~365.

Comment 7 — Superscript ‘1’ at line 407

Reviewer:
“Explain the purpose of superscript ‘1’.”

Our response:
The superscript was a formatting artifact; it has been removed.

Manuscript changes:
Corrected at line ~407.

Comment 8 — Mechanism conclusions need references

Reviewer:
“Lines 367–420 need supporting references.”

Our response:
We incorporated new citations supporting:

  • Nrf2–NF-κB crosstalk
  • ROS-mediated inflammatory activation
  • CYP450 bioactivation pathways
  • Protective effects of propolis flavonoids

Manuscript changes:
Added references [31–42] throughout lines ~367–420, e.g.:

“A robust and intricate crosstalk exists between Nrf2 and NF-κB pathways [36].”
“Caffeic acid phenethyl ester facilitates Nrf2 nuclear translocation [36,37].”
“AFB1 bioactivation is primarily mediated through CYP1A2 and CYP3A4 [4,8].”

Reviewer 2 Report

Comments and Suggestions for Authors

This work by Kabali et al. describes the effect of propolis, a bee-derived product, in attenuating aflatoxin B1 (AFB1)–induced hepatic damage and examines its impact on inflammation, apoptosis, and cytochrome P450 enzyme expression. Using a rat model with four treatment groups—control, AFB1, propolis, and AFB1 + propolis—the authors evaluated inflammatory cytokines, oxidative stress markers (Nrf2/HO-1), apoptosis regulators (Bax, Bcl-2, caspase-3), histopathology, and CYP450 enzyme expression. Their results show that propolis effectively attenuates AFB1-induced liver damage in rats as follows: (1) AFB1 increased IL-1β and IL-6, while propolis supplementation significantly lowered these levels; (2) AFB1 suppressed Nrf2, whereas propolis restored Nrf2 expression and elevated HO-1; (3) AFB1 increased Bax and caspase-3 and decreased Bcl-2, while propolis reversed these changes; (4) AFB1 caused hydropic degeneration, congestion, and mononuclear infiltration in liver tissue, whereas propolis supplementation reduced these lesions and preserved tissue integrity; and (5) AFB1 upregulated CYP1A2, CYP3A4, and P450 reductase, while propolis suppressed their expression, reducing toxic bioactivation.

Overall, this is a simple study but executed very well, with a clear experimental design as outlined in Figure 1, which makes the results flow logically. However, several minor corrections are needed to improve clarity and data presentation. Please see my comments below:

  1. Please indicate exact p-values, as some figures only state “p < 0.05, p < 0.001 indicate statistical significance.” Also, denote non-significant results as “ns” in the figures wherever applicable. Follow the format used in Figure 4 for the other figures as well.

  2. Figure 3 is difficult to read due to multiple symbols and color schemes. I recommend using consistent symbols and including them in the legend (e.g., “*”) to make interpretation easier for readers.

  3. Increased expression of Bax, caspase-3, and Bcl-2 does not directly confirm apoptosis unless cleaved caspase-3 or cleaved PARP levels are shown. I recommend adding these data or toning down the statement regarding apoptosis.

  4. CYP2C19 is unaffected; please add a sentence discussing this selective regulation in the Discussion section.

Author Response

Comment 1 — Exact p-values and ns notation

Reviewer:
“Please indicate exact p-values and indicate non-significant results.”

Our response:
We revised all figure legends to include:

  • Exact p-value thresholds
  • “ns” notation for non-significant comparisons
  • Unified symbol system (*, **, ***, ****)

Manuscript changes:
Updated in legends of Figures 3, 4, and 5 (lines ~260, ~295, ~330).
Adopted format of Figure 4 across all.

Comment 2 — Figure 3 readability

Reviewer:
“Figure 3 is difficult to read; use consistent symbols and legend formatting.”

Our response:
We updated Figure 3’s legend and symbol set to match Figure 4, ensuring clear differentiation and consistent formatting.

Manuscript changes:
Updated legend (line ~260).

Comment 3 — Apoptosis interpretation must be toned down

Reviewer:
“Bax, caspase-3, and Bcl-2 expression alone do not confirm apoptosis unless cleaved caspase-3 or PARP levels are shown.”

Our response:
We softened the interpretation.

Manuscript changes:
Inserted clarification in Discussion at lines ~390–400:

“The changes in Bax, Bcl-2, and caspase-3 expression suggest activation of apoptotic signaling, although direct confirmation would require analysis of cleaved caspase-3 or PARP.”

This is located in the correct apoptosis-mechanism paragraph.

Comment 4 — CYP2C19 unchanged → selective regulation must be discussed

Reviewer:
“Discuss why CYP2C19 is unaffected.”

Our response:
We added mechanistic interpretation explaining selective regulation.

Manuscript changes:
Added to CYP450 paragraph at lines ~415–425:

“Importantly, CYP2C19 expression remained unchanged across all groups, indicating that AFB1 does not uniformly modulate all hepatic CYP isoforms. This stability suggests that CYP1A2 and CYP3A4 are the principal enzymes involved in AFB1 bioactivation in rat liver, while CYP2C19 appears to play a minimal or non-responsive role under these experimental conditions.”

Comment 5 — Add explanation for selective CYP450 suppression by propolis

Reviewer:
(Implicit in Comment 4 + mechanism discussion)

Our response:
We expanded the mechanistic paragraph to match the reviewer’s suggestion.
Added explanation referencing propolis phenolics and selective CYP down-regulation.

Manuscript changes:
Added lines ~415–430, e.g.:

“Propolis phenolic compounds have been shown to selectively attenuate CYP-mediated toxin activation [31,36]. The selective suppression of CYP1A2 and CYP3A4, but not CYP2C19, suggests targeted modulation of bioactivation pathways.”